# Importance of Initial Interfacial Steps during Chalcopyrite Bioleaching by a Thermoacidophilic Archaeon

**DOI:** 10.3390/microorganisms8071009

**Published:** 2020-07-06

**Authors:** Camila Safar, Camila Castro, Edgardo Donati

**Affiliations:** CINDEFI (CCT La Plata —CONICET, U.N.L.P.), Facultad de Ciencias Exactas, Universidad Nacional de La Plata, La Plata 1900, Argentina; camila.safar@gmail.com

**Keywords:** thermophiles, bioleaching, chalcopyrite, interface interactions, metals, microbial adhesion, archaea

## Abstract

Studies of thermophilic microorganisms have shown that they have a considerable biotechnological potential due to their optimum growth and metabolism at high temperatures. Thermophilic archaea have unique characteristics with important biotechnological applications; many of these species could be used in bioleaching processes to recover valuable metals from mineral ores. Particularly, bioleaching at high temperatures using thermoacidophilic microorganisms can greatly improve metal solubilization from refractory mineral species such as chalcopyrite (CuFeS_2_), one of the most abundant and widespread copper-bearing minerals. Interfacial processes such as early cell adhesion, biofilm development, and the formation of passive layers on the mineral surface play important roles in the initial steps of bioleaching processes. The present work focused on the investigation of different bioleaching conditions using the thermoacidophilic archaeon *Acidianus copahuensis* DSM 29038 to elucidate which steps are pivotal during the chalcopyrite bioleaching. Fluorescent in situ hybridization (FISH) and confocal laser scanning microscopy (CLSM) were used to visualize the microorganism–mineral interaction. Results showed that up to 85% of copper recovery from chalcopyrite could be achieved using *A. copahuensis*. Improvements in these yields are intimately related to an early contact between cells and the mineral surface. On the other hand, surface coverage by inactivated cells as well as precipitates significantly reduced copper recoveries.

## 1. Introduction

Thermophiles have been found to be widespread in numerous ecosystems, including volcanic environments, hot springs, mud pots, fumaroles, geysers, thermal springs, and even deep-sea hydrothermal vents [1]. Studies of thermophilic microorganisms have shown that they have a considerable biotechnological potential due to their optimum growth and metabolism at high temperatures [2,3,4]. The importance of expanding our knowledge on microorganisms derived from extreme environments stems from the development of novel and sustainable technologies for our health, environment, and food industry.

Thermophilic archaea have been studied to a limited extent, compared to mesophilic microorganisms. Archaea are peculiar in various aspects; they have unique biochemical and physiological characteristics with important biotechnological applications [4,5]. Many of these archaeal species, which tolerate conditions with low pH, high temperatures and concentration of metal ions, could be used in bioleaching processes to recover valuable metals from mineral ores. These microorganisms can oxidize ferrous iron and/or reduced inorganic sulfur compounds (RISCs), thus generating ferric iron and/or protons that could catalyze the dissolution of metal sulfides [6]. Bench scale experiments have demonstrated that thermophiles improve copper recoveries from primary copper sulfides, such as chalcopyrite (CuFeS_2_), enargite (Cu_3_AsS_4_), and covellite (CuS) [7,8,9]. Also, these microorganisms were successfully employed to recover other metals such as nickel, molybdenum, uranium, and zinc from low-grade and polymetallic ores, and in the biooxidation of refractory gold-bearing ores [10,11,12,13]. Pilot scale bioleaching processes using thermophiles have shown to be successful processes to recover metals from Chuquicamata Mine (Chile) and Aguablanca Mine (Spain) [14,15]. The use of thermophiles in bioleaching has become the spotlight in recent years since the use of higher operational temperatures would eliminate the need of energy input for cooling the system and, additionally, decrease the passivation of mineral surfaces [16]. Thus, thermoacidophilic microorganisms can greatly improve metal solubilization from refractory mineral species such as chalcopyrite, one of the most abundant and widespread copper-bearing minerals.

Bioleaching has some advantages over conventional mining techniques, such as low operation costs, low investment in infrastructure, reduced emissions to the air, simplicity of operation, and applicability to refractory ores and minerals that cannot be treated by traditional techniques. It is a complex process involving several chemical and physical factors. Amongst them, interfacial phenomena such as cell adhesion to the mineral surface and biofilm development are of great importance for the mineral dissolution. The attachment of microorganisms to the mineral surface is relevant in the bioleaching process since the leaching reaction mostly occurs in the microorganism–mineral interface [17]. Extracellular polymeric substances (EPS), mainly composed of carbohydrates, proteins, lipids, nucleic acids, and complexed metal ions, are involved in the attachment to mineral surfaces and the later formation of biofilms [18]. Several studies have provided significant knowledge about these processes for mesophilic microorganisms [19,20]. Recently, studies on the adhesion to mineral substrates, EPS production, and biofilm formation by thermoacidophilic archaea such as the genera *Acidianus* and *Sulfolobus* have been reported [21,22,23,24]. However, only a few of them refer to chalcopyrite. Therefore, further understanding of adhesion and biofilm formation of thermoacidophilic archaea on chalcopyrite surface could offer valuable information to the biomining industry in order to improve copper recovery from the still very abundant chalcopyrite resources.

Since thermophilic microorganisms are efficient in the bioleaching of refractory minerals, in this work the bioleaching of a chalcopyrite concentrate was evaluated using the thermoacidophilic archaeon *Acidianus copahuensis* DSM 29038. The adhesion of these microorganisms to the mineral surface, as occurs with mesophiles, could be an important stage that determines the speed and efficiency of the bioleaching process. For this reason, different conditions were tested to elucidate the pivotal steps during the chalcopyrite bioleaching. In addition, the colonization of chalcopyrite surface by *A. copahuensis* was visualized using fluorescent in situ hybridization (FISH) and confocal laser scanning microscopy (CLSM). This information has great importance for the biomining industry to offer strategies to improve metal recoveries by bioleaching.

## 2. Materials and Methods

### 2.1. Strain and Culture Conditions

*A. copahuensis* strain ALE1 DSM 29038 [25] was cultivated in flasks containing Mackintosh basal salt solution (MAC) [26] at pH 2 supplemented with 10 g/L elemental sulfur (S°) powder and 1 g/L yeast extract as energy source. Cultures were incubated in Erlenmeyer flasks containing 100 mL of fresh medium at 65 °C with agitation at 150 rpm.

### 2.2. Mineral

Mineral samples from Alumbrera Mine (Catamarca, Argentina) were used throughout this study. The main chemical composition of the concentrate is (w/w): 24.20% Cu, 31.15% Fe, 0.76% Mo, 0.56% Zn, 0.26% Pb, 0.02% Ag, 0.002% Ni, and 0.002% Au. The main mineralogical species detected in the mineral were chalcopyrite (77.4%), pyrite (19.6%), molybdenite (2%), and sphalerite (< 1%). The fraction with particle size < 62 μm was used in the experiments. The specific surface area of the mineral was 5.36 m^2^/g (BET surface area).

### 2.3. Bioleaching Experiments

The bioleaching assay was performed in Erlenmeyer flasks containing 150 mL of MAC at pH 2 supplemented with 10 g/L mineral. Inoculation was performed using enough volume of a pellet resuspended in fresh medium to reach 1 × 10^8^ cells/mL in the flasks. The pellet was prepared by centrifugation of a filtered growth culture (<2 µm pore to eliminate insoluble sulfur) in the same conditions as described in Section 2.1. Flasks were incubated at 65 °C with agitation at 150 rpm for 42 days. Different conditions were tested in triplicate, including (Figure 1): (1) inoculation at the beginning of the assay (t_0_); (2) inoculation a day before t_0_ and replacement of the liquid medium at t_0_, maintaining the pre-colonized mineral; (3) inoculation a day before t_0_, inactivation of microorganisms at t_0_ by exposure for 1 h at 100 °C, followed by a replacement of the liquid medium and re-inoculation; (4) the same conditions as in (3) but without re-inoculation, and (5) replacement of inoculum by an addition of the same volume of p-formaldehyde (PFA) at the beginning of the assay (sterile control).

Samples were taken periodically over 42 days. Before the procedure, sterile water was added to the flasks to replace water lost by evaporation. For this purpose, the weight of each flask was recorded after taking each sample. Approximately 4 mL of different bioleaching cultures was filtered to remove the mineral and perform several analyses. The pH and redox potential (ORP) were measured using specific electrodes, total Cu and Fe concentrations in solution were quantified by atomic absorption spectroscopy (AAS), and ferrous iron concentration was determined using a colorimetric method with an *o*-phenantroline reaction [27].

### 2.4. Fluorescent In Situ Hybridization

Fluorescent in situ hybridization was performed on PFA-fixed mineral samples [28]. Hybridizations were done following Amann’s protocol [29] using Cy3 labeled probes: ARCH915 (5′-GTGCTCCCCCGCCAATTCCT-3′, 20% formamide in hybridization buffer) specific for Archaea domain [30]. After hybridization, samples were stained with SYTO 62 (Molecular Probes). Dako Fluorescent Mounting Medium (Dako North America Inc., USA) was used in order to avoid fluorescence fading. Samples were visualized by confocal laser scanning microscopy using a Leica TCS SP5 microscope, and images were processed using ImageJ software [31].

### 2.5. EPS Staining

PFA-fixed mineral samples were stained to visualize EPS production of attached cells. TRITC- labeled Concanavalin A (Invitrogen) was used to stain EPS by binding with α-D-mannose and α-D-glucose residues according to previous studies [32]. SYTO 62 was employed to stain cells. Dako Fluorescent Mounting Medium was used in order to avoid fluorescence fading. Microscopy and image processing were done as described in Section 2.4.

## 3. Results and Discussion

### 3.1. Adhesion to Chalcopyrite Surface

Microscopic images are shown in Figure 2. Initially, the ore particles were similar in shape, color and size, and the surface of the grains seemed to be clean and smooth; the ore grains had only a few cracks and imperfections. Over the course of the incubation time, the ore grains became more corroded due to the leaching process, and irregular edges appeared. After 20 days of incubation, some particles with different color and size could be visualized (data not shown); these particles seemed to be precipitates formed during the leaching process, probably some insoluble sulfides and polysulfides, elemental sulfur and/or insoluble sulfates [33]. At the end of the experiment, the mineral grains presented a high degree of corrosion with pits distributed over the entire mineral surface.

The ARCH915 probe combined with SYTO 62 was used to determine the presence of *A. copahuensis* cells attached to chalcopyrite grains. The SYTO 62 signal (red) allowed the detection of the entire cell population attached to the mineral, regardless of its metabolic state. While the signal from the ARCH915 probe (green) allowed the detection of achaeal cells with a high number of ribosomes, which is to be expected in metabolically active cells. Sessile cells were detected at all stages of the bioleaching process, from the beginning to the end of the experiment (Figure 2). Sessile cells were heterogeneously distributed on the surface, located mainly near cracks and imperfections of the mineral. During the first days, the SYTO 62 signal colocalized with the ARCH915 probe signal. This result indicates that initially the sessile population was mostly metabolically active. However, a small population of inactive sessile cells was also detected, suggesting that this small population could play an important role in the structure of the biofilm, even in the early stages of its development.

Glycoconjugates from EPS mediating the contact between cells and minerals were detected through the use of the fluorescently labeled lectin ConA-TRITC (Figure 3); a phenomenon also reported for other acidophilic biofilms [34,35,36]. The EPS could play different functions such as adhesion to surfaces, protective barrier, water retention, reservoir of nutrients, and enzymatic activity, among others [18,37]. It is interesting to note that some areas with ConA-TRITC signal were devoid of SYTO 62 signal, suggesting EPS were concentrated on surrounding cells but also along the surface interconnecting different cell clusters. Similarly, Schofp et al. [38] reported that EPS mediates the contact between different archaeal species in a mixed biofilm, which could facilitate the communication and cooperation through diffusion of metabolites between cell clusters. Also, these microbial footprints were detected in the interaction of *Acidianus spp*. and *Sulfolobus metallicus* with pyrite [21]. These results reveal that detachment processes play an important role during biofilm formation of archaeal species on mineral surfaces such as pyrite and chalcopyrite.

After 20 days of incubation, fluorescent signals were stronger and thoroughly spread over the mineral surface compared to grains after 1 day of incubation (Figure 2b). At this time, mineral particles exhibited some areas stained with SYTO 62 but not hybridized with the probe ARCH915, indicating that part of the sessile population was inactive or dead. At the end of the experiment, FISH hybridization showed that some cells remained attached to the mineral surface (Figure 2c). However, ARCH915 signal was localized only in a few areas, indicating an increment of the inactive population in the biofilm. These results could be attributed to different events that may occur in later phases of the experiment, including nutrient depletion, formation of passive barriers that limit the diffusion of nutrients into cells, and accumulation of metals or metabolic products that may inhibit the microbial activity and reduce the viability [37].

### 3.2. Chalcopyrite Bioleaching

Different bioleaching conditions were tested in order to study the effects of the mineral surface coverage by inactive cells.

It can be seen in Figure 4 that in both abiotic systems (conditions 4 and 5) there was a rapid increase in pH. This is due to the fact that initially in abiotic systems the dissolution of chalcopyrite was only achieved by a non-oxidative reaction mediated by protons, which releases ferrous iron, copper, and hydrogen sulfide [33]:

CuFeS_2_ + 4 H^+^ → Fe^2+^ + Cu^2+^ + 2 H_2_S,
(1)

The slight delay in the pre-colonized abiotic system (condition 4) may be due to the partial coverage of the mineral surface during the pre-colonization and cell inactivation steps before the assay started. This coverage, generated by the cells adhered to the mineral surface and precipitates, prevented further extension of the acid dissolution mechanism.

Figure 5 shows the variation of ORP during bioleaching tests. ORP values were mainly governed by the ratio of ferric to ferrous iron [33]. Low ORP values were recorded in abiotic conditions. This is because the dissolution of chalcopyrite released ferrous iron (Equation (1)), and its chemical oxidation was very slow at such low pH values. Comparing both abiotic controls, a high baseline ORP was observed in condition 4, probably due to the ferrous iron oxidation catalyzed by *A. copahuensis* cells during the pre-colonization step, as long as cells were active. Thus, the increase in ferrous iron concentration reduced the ORP values close to those values reached by the not pre-colonized and non-inoculated system (condition 5).

It should be noted that the delay in the pre-colonized abiotic system (condition 4) with respect to the non-inoculated system (condition 5) was also evidenced in the evolution of ferrous iron and total iron concentrations during the experiment (Figure 6). Likewise, a similar trend in copper leaching was observed in both abiotic systems (Figure 7).

In abiotic systems, during the first days of the experiment, the copper/iron molar ratios in solution were close to 1, which fits an acidic dissolution stoichiometry. Then, a rapid decrease in the copper/iron molar ratios was observed, reaching values around 0.2. This is because the accumulation of ferrous iron and copper in solution under low ORP conditions enabled a second chalcopyrite dissolution mechanism, known as reductive mechanism [39]:

CuFeS_2_ + Fe^2+^ + Cu^2+^ + 2 H^+^ → Cu_2_S + 2 Fe^3+^ + H_2_S,
(2)

In these systems, the copper/iron molar ratios reached values around 0.3 at the end of the experiment. Once enough ferric iron has accumulated in the leaching solution, chalcopyrite dissolution may occur by an oxidative mechanism, represented by the following reactions [39]:

CuFeS_2_ + 4 Fe^3+^ → Cu^2+^ + 5 Fe^2+^ + 2 S^0^,
(3)

2 Fe^3+^ + H_2_S → 2 Fe^2+^ + S^0^ + 2 H^+^,
(4)

Cu_2_S + 4 Fe^3+^ → 4 Fe^2+^ + S^0^ + 2 Cu^2+^,
(5)

These processes, which occur at low levels, contributed to a greater increase in the production of copper moles than iron moles. This was reflected by an increase in the copper/iron molar ratio towards the end of the experiment.

In inoculated systems, the evolution of pH was clearly different from that in abiotic controls. Except for the first days, the pH began to drop similarly in all biotic systems (Figure 4), suggesting that the reductive mechanism (Equation (2)), which consumes protons, was not relevant except at the very beginning of the experiment. In the initial stage, there was an increase in the pH value due to the reductive mechanism but also to iron oxidation catalyzed by microorganisms [17]:

2 Fe^+2^ + ½ O_2_ + 2 H^+^ → 2 Fe^+3^ + H_2_O,
(6)

The presence of ferric iron activated the oxidative mechanism (Equations (3)–(5)) and generated elemental sulfur that could also be biologically oxidized [17]:

S^0^ + ³/₂ O_2_ + H_2_O → SO_4_^−2^ + 2 H^+^,
(7)

In addition, the hydrolysis of ferric iron, which is significantly increased by temperature, also leads to a decrease in pH. This hydrolysis can result in the production of different solid phases such as goethite and schwertmannite, as shown by Equations (8) and (9), respectively [17,33,40]. However, the pH and redox potential conditions in inoculated systems favored jarosite production (Equation (10)) [40]:

Fe^3+^ + 3 H_2_O → Fe(OH)_3_(s) + 3 H^+^,
(8)

8 Fe^3+^ + SO_4_^2−^ +14 H_2_O → Fe_8_O_8_(OH)_6_SO_4_(s) + 22 H^+^,
(9)

K^+^ + 3 Fe^3+^ + 2 SO_4_^2−^ + 6 H_2_O → KFe_3_(SO_4_)_2_(OH)_6_(s) + 6 H^+^,
(10)

The evolution of ORP in inoculated systems (conditions 1, 2, and 3) tended to increase over the course of the experiment, reaching values between 550 and 600 mV (Figure 5). Comparing these systems, those pre-colonized (conditions 2 and 3) exhibited a high initial ORP value. Then, ORP started to decrease, reaching similar values to that recorded in the not pre-colonized system (condition 1). Finally, all inoculated cultures exhibited an ORP increase. The differences in the initial ORP values could be attributed to the fact that during the pre-colonization period the initial acid attack on the chalcopyrite generated a small but significant amount of ferrous iron. Its oxidation generated ferric iron that could be adsorbed and/or associated with sessile cells. This ferric iron was partially released into solution when the culture medium was replaced at the start of the assay, generating a higher initial ORP value in the pre-colonized systems compared to the not pre-colonized one.

In Figure 6, it can be seen that initially, ferrous iron concentration increased in all inoculated systems, probably generated by acidic attack (Equation (1)). Its fast oxidation by microbial catalysis (Equation (6)) maintained the ferrous iron concentration negligible throughout the experiment.

Copper dissolution was clearly higher in condition 2, achieving approximately 85% after 30 days of incubation (Figure 7). Under conditions 1 and 3, about 60% of total copper dissolution was reached at the end of the experiment. However, the initial copper dissolution kinetics for condition 3 were much slower than for the other two conditions. Surely, it was connected with the initial coverage of the surface by inactive cells, and also perhaps by oxidized phases generated during the heat inactivation treatment (a phenomenon also observed in condition 4).

The copper/iron molar ratios in inoculated systems fluctuated in the ranges 1.5–3.5, 1.2–4.5, and 1.5–7.3 for condition 1, 2, and 3, respectively. These ratios were quite a bit higher than the expected values according to the stoichoimetric coefficients in chalcopyrite. The reductive mechanism (Equation (2)) tends to lower such ratios even below the stoichiometric ratio by forming a solid phase of copper. That process is favored just at low ORP; this is why its participation in chalcopyrite dissolution operating in the cultures was not relevant. On the other hand, the abundant precipitation was responsible for such high molar ratio values in the leachates.

These results show that the mechanism of chalcopyrite dissolution was essentially the oxidative one (Equations (3)–(5)) in the three biotic systems. The replacement of the culture medium (condition 2) delayed ferric iron increase. This, consequently, also delayed its hydrolysis and precipitation (Equations (8)–(10)). Thus, condition 2 favored the mineral colonization by *A. copahuensis* before significant precipitation of ferric iron compounds and coverage of the mineral surface occurred. This fact was reflected by a higher leaching efficiency in this culture. Conversely, ferric iron precipitation occurred from very early stages in systems without medium replacement (conditions 1 and 3). This precipitation generated a solid layer of ferric iron phases on the mineral surface. The progressive coverage of the mineral surface caused diffusion problems, making chalcopyrite dissolution difficult. This coverage also prevented, or at least hindered, the adhesion of cells on the mineral surface. Thus, in systems without culture medium replacement, the contact mechanism was prevented. However, the slow kinetics of initial copper dissolution in condition 3 indicate that there was a major surface impediment that limited the contact between *A. copahuensis* cells and the mineral. Probably, the coverage with cells later inactivated (during the pre-colonization step) partially blocked the mineral surface.

In order to support the last conclusions, the rate-determining step in copper solubilization, bioleaching kinetics were analyzed by applying the shrinking core model [41,42,43]. The following equation of the shrinking core model describing the kinetics can be used:

K_p_t = 1 − ⅔x − (1 − x)^⅔^,
(11)

K_p_t = 1 − (1 − x)^⅓^,
(12)
where K_p_ is the parabolic rate constant (d^−1^), t is time (d), and *x* is the mole fraction of reacted copper.

Plots of 1 − ⅔x − (1 − x)⅔ versus time are shown in Figure 8a for the experimental results reported here for the bioleaching conditions. Plots 1 − (1 − x)⅓ versus time are presented in Figure 8b using experimental values.

Taking into account the correlation coefficients (R^2^), results showed that conditions 1 and 2 fit better with Equation (11) (R^2^ were 0.993 and 0.999, respectively) than with Equation (12) (R^2^ were 0.970 and 0.967, respectively). This indicates that the model of copper dissolution kinetics controlled by diffusion through the product layer correlates much better than the kinetic model limited by chemical reaction. These results are consistent with our previous discussion, suggesting the existence of a product layer on the mineral surface (probably jarosite) that controlled the bioleaching of chalcopyrite under these conditions. It can also be observed that the linear trend of condition 1 started earlier than condition 2, indicating, as we suggested above, that the replacement of the culture medium delayed the precipitation of ferric iron compounds, thus preventing the formation of a product layer on the mineral surface during early stages.

In contrast, condition 3 fits better with a chemical reaction-controlled model compared to the diffusion model (correlation coefficients of 0.949 and 0.894, respectively). However, it should be noted that the correlation coefficients for both dissolution models were not substantially suitable to ensure that one model prevailed over the other. This result could be interpreted assuming that chalcopyrite bioleaching was limited by the presence of inactive cells that restricted the contact between active cells and the mineral surface.

This experiment has shown that up to 85% of copper recovery from the chalcopyrite concentrate could be achieved using *A. copahuensis*, confirming that thermoacidophilic microorganisms could be successfully used in a bioleaching process with refractory minerals. The maximum extraction achieved in this experience was clearly greater than that achieved using mesophilic microorganisms, usually less than 40–50% [44]. The fundamental role of the microbial contact mechanism and the importance of surface colonization in the process of chalcopyrite bioleaching were also demonstrated. It was found that the partial blocking of the surface, both by inactive cells and by deposition of solid phases, decreases bioleaching rates and final copper extractions.

In biotic systems, the formation of ferric iron phases progressively reduced the diffusion of reagents into viable cells attached to the surface. Under these conditions, the diffusion of reagents through the product layer regulated the kinetics of the bioleaching process. The particular design of the experiment reported in this work allowed a greater colonization of the mineral surface before the beginning of the precipitation of ferric iron compounds, and a slower increase of ORP that delayed the precipitation of solid ferric compounds on the surface. In this way, a greater extension of the microbial contact mechanism was achieved and consequently, high copper extraction was obtained.

## 4. Conclusions

In the present study, the ability of *A. copahuensis* cells to colonize and form biofilms on the chalcopyrite surface was explored. This species efficiently contributed through an oxidative mechanism to chalcopyrite dissolution with high copper recovery. To guarantee the continuity of the bioleaching process, it is essential to maintain low precipitation of ferrous compounds mainly in the early stages, allowing a greater extension of microbial contact. Our results suggest that initial adhesion with low microbial activity to maintain low values of ORP is a good strategy to achieve that, and consequently, to reach high copper recoveries.

## Figures and Tables

**Figure 1 microorganisms-08-01009-f001:**
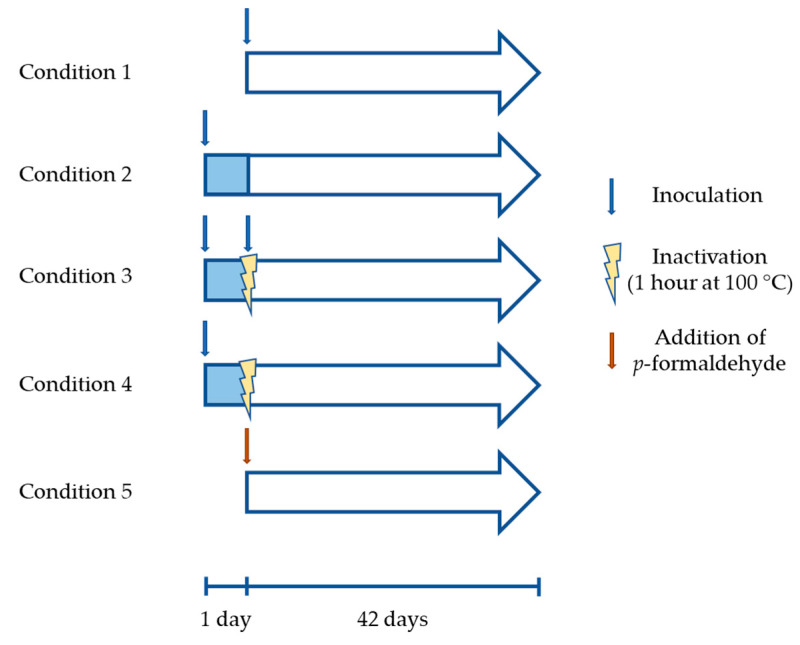
Scheme of the different bioleaching conditions tested.

**Figure 2 microorganisms-08-01009-f002:**
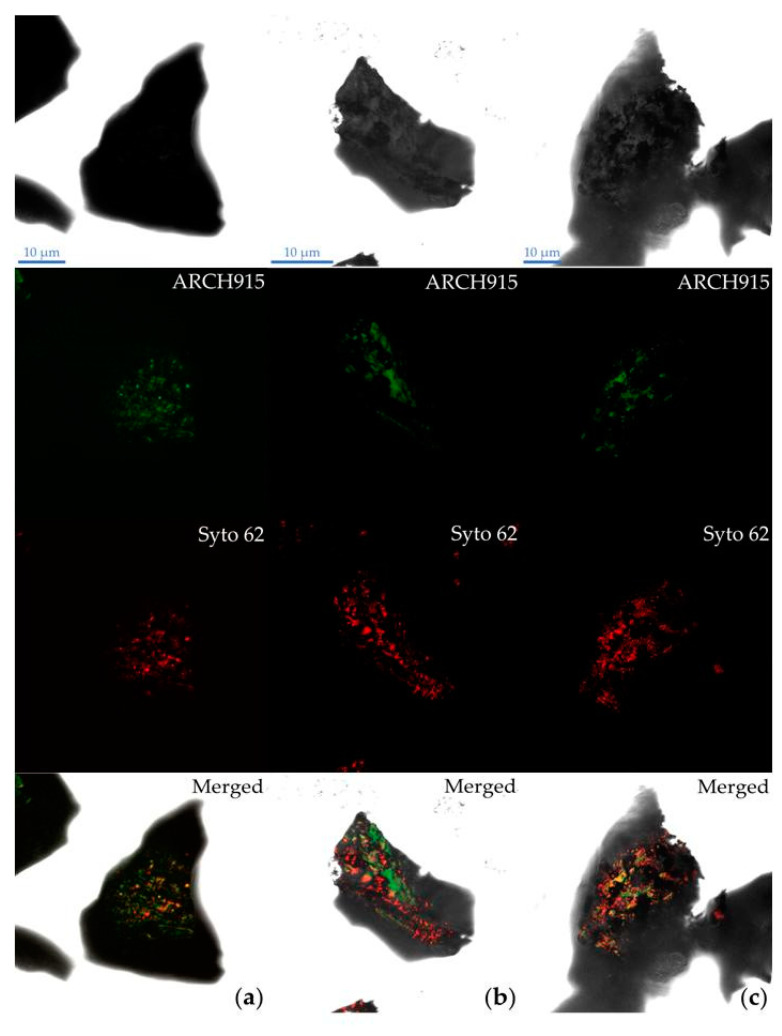
*A. copahuensis* biofilm development on chalcopyrite surface detected by FISH using ARCH915 probe (green) and SYTO 62 (red) at: (**a**) 1 day; (**b**) 20 days; (**c**) 42 days. Scale bars represent 10 µm.

**Figure 3 microorganisms-08-01009-f003:**
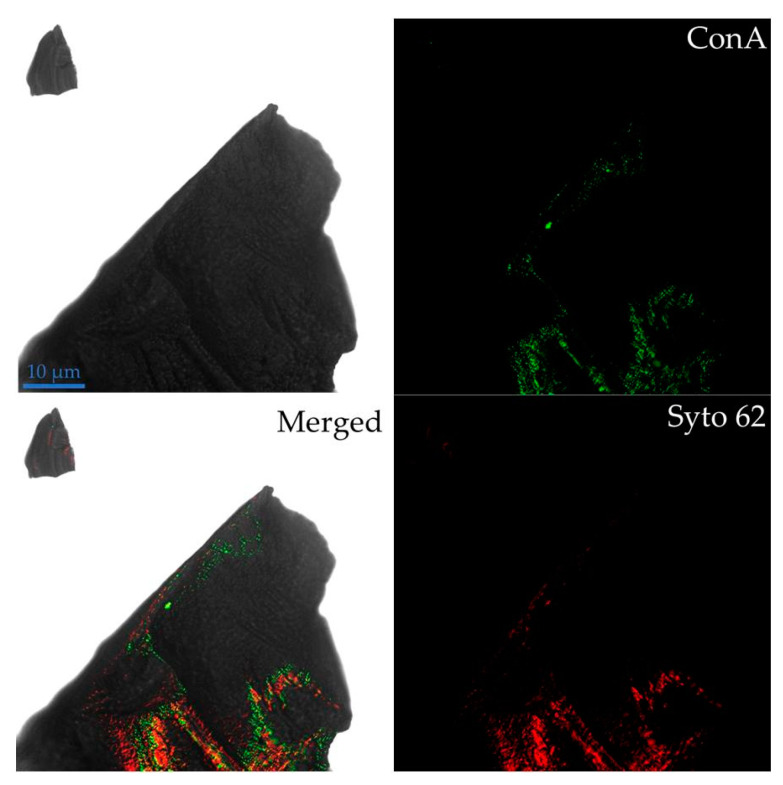
*A. copahuensis* cells attached to a chalcopyrite grain visualized by CLSM; cells were stained with SYTO 62 (red) and ConA specifically bound to EPS. Scale bar represents 10 µm.

**Figure 4 microorganisms-08-01009-f004:**
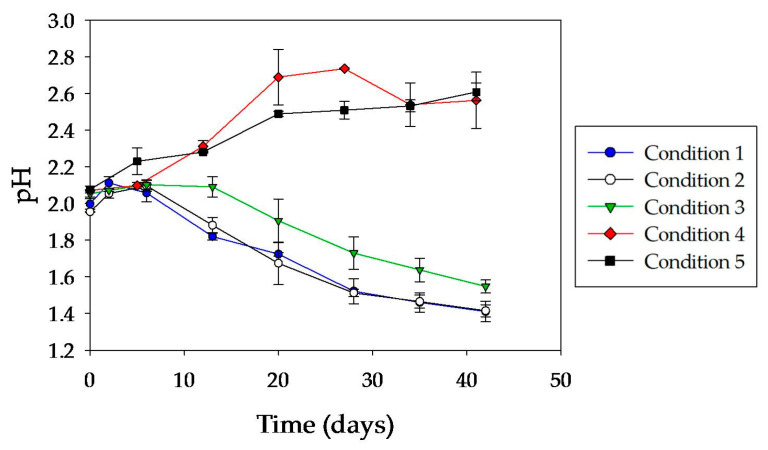
Evolution of pH during chalcopyrite leaching tests with *A. copahuensis.* Chalcopyrite dissolution was tested in shake flask cultures with 2% (w/v) pulp density incubated at 65 °C for 42 days. Error bars represent standard deviation from three independent experiments (n = 3).

**Figure 5 microorganisms-08-01009-f005:**
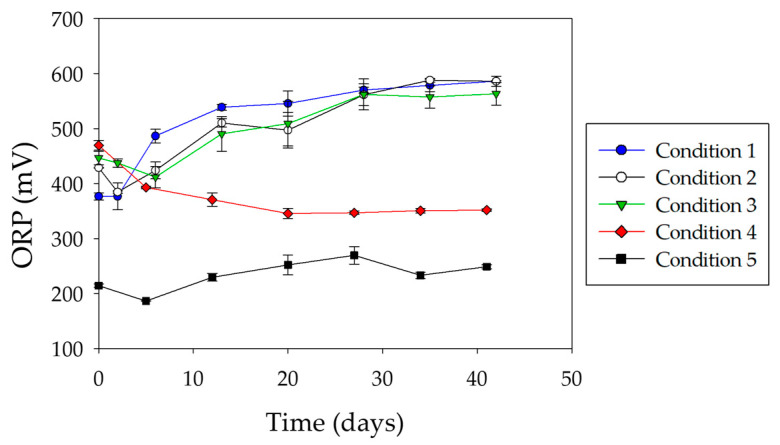
Evolution of ORP during chalcopyrite leaching tests with *A. copahuensis*. Chalcopyrite dissolution was tested in shake flask cultures with 2% (w/v) pulp density incubated at 65 °C for 42 days. Error bars represent standard deviation from three independent experiments (n = 3).

**Figure 6 microorganisms-08-01009-f006:**
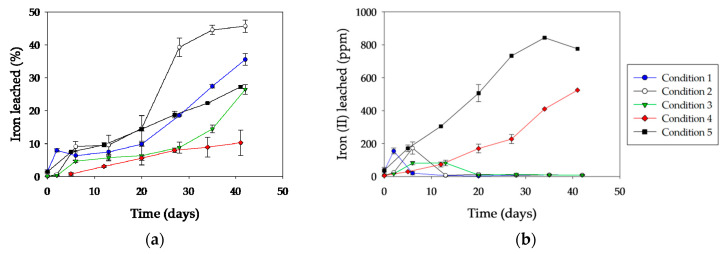
Evolution of iron concentration during chalcopyrite leaching tests with *A. copahuensis*: (**a**) total iron dissolutions; (**b**) ferrous iron concentrations. Chalcopyrite dissolution was tested in shake flask cultures with 2% (w/v) pulp density incubated at 65 °C for 42 days. Error bars represent standard deviation from three independent experiments (n = 3).

**Figure 7 microorganisms-08-01009-f007:**
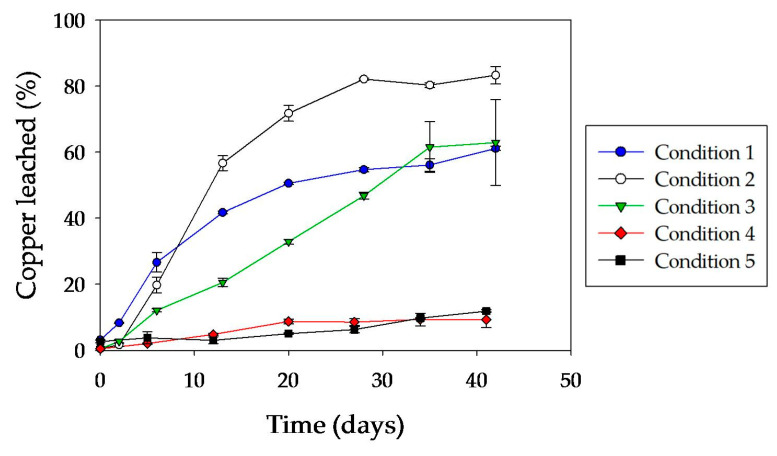
Kinetics of copper solubilization from a chalcopyrite concentrate by *A. copahuensis.* Chalcopyrite dissolution was tested in shake flask cultures with 2% (w/v) pulp density incubated at 65 °C for 42 days. Error bars represent standard deviation from three independent experiments (n = 3).

**Figure 8 microorganisms-08-01009-f008:**
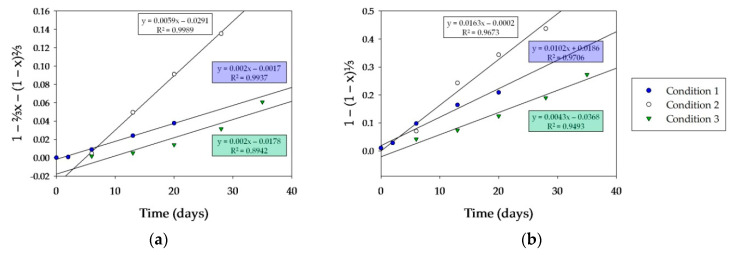
Kinetic modeling of chalcopyrite bioleaching: (**a**) product layer diffusion-controlled model; (**b**) chemical reaction-controlled model, based on data shown in Figure 7.

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
