# Peer review of "Importance of Initial Interfacial Steps during Chalcopyrite Bioleaching by a Thermoacidophilic Archaeon"

_microorganisms, 2020, doi:10.3390/microorganisms8071009_

Round 1

Reviewer 1 Report

The manuscript describes bioleaching of chalcopyrite by using a thermophilic archaeon. Different approaches were applied for the experiments, which differed in the initial conditions, especially for the cell attachment, that influenced the microbial activity. Principally, the chalcopyrite dissulotion by various microorganisms and influence of redox potential and secondary mineral formation on bioleaching of chalcopyrite has been studied many times. Howecer, the present study investigate the importance of conditions for intitial cell attachement and its influence on bioleaching activity very well, reaching a remarkable high Cu release of 85 %. Experimental set-up and data interpretation is well described and comprehensive. Therefore I recommend the article for publication.

Minor comment: there are some typos that should be corrected, e.g.

Acidianus copahuensis should be written in italic letters throughout the text

  1. 157: enzymatic

l. 180: different

Author Response

We are very glad to submit the revised and corrected version of the manuscript Importance of initial interfacial steps during chalcopyrite bioleaching by a thermoacidophilic archaeon" to be considered for publication in Microorganisms. We would like to thank you for the thorough reading of the manuscript and their suggestions that contributed to improve its quality. In the corrected version we have addressed all the suggestions and comments (the modifications were highlighted in yellow and were made using the “track changes” function of MicrosoftWord). Please see the detailed responses to Reviewer #1 in the attached file.

Reviewer 2 Report

The information presented in the paper is very important, and article reports new evidence concerning mechanisms of chalcopyrite bioleaching as a result of biofilm formation by thermoacidophilic archaeon Acidianus copahuensis. Especially important is the authors' suggestion that initial adhesion of cells with low metabolic activity, which leads to maintain low values of redox potential is a good strategy to reach high copper recoveries. The paper is well written and merits publication.

Some minor points for consideration:

  1. Use italic for name of species (Acidianus copahuensis, A. copahuensis).
  2. Line 151: Figure 2. copahuensis biofilm development in on chalcopyrite surface…
  3. Eqs 1-9: The chemical reactions represented by Eqs 1-9 require reference to literature sources, as the authors did not carry out analysis of ferric iron concentration as well as XRD and XPS studies to analyze all substances (both substrates and products) that appear in these equations. Only Eq. 10 is correctly described.

Author Response

We are very glad to submit the revised and corrected version of the manuscript Importance of initial interfacial steps during chalcopyrite bioleaching by a thermoacidophilic archaeon" to be considered for publication in Microorganisms. We would like to thank you for the thorough reading of the manuscript and their suggestions that contributed to improve its quality. In the corrected version we have addressed all the suggestions and comments (the modifications were highlighted in yellow and were made using the “track changes” function of MicrosoftWord). Please see the detailed responses to Reviewer #2 in the attached file.

Reviewer 3 Report

Demand for copper is increasing, and therefore improving yields of recalcitrant chalcopyrite bioleaching is highly desired. Passivation layers during conventional bioleaching lower copper extraction rates. Bioleaching at increased temperatures (65 to 80 °C) can circumvent this problem. The authors present an interesting study on bioleaching of chalcopyrite by the thermoacidophilic archaeon Acidianus (A.) copahuensis. The sulphur- and iron-oxidizing A. copahuensis has been described recently, and data on its bioleaching capacity are therefore scarce in literature. Kinetic data are presented in this study and bioleaching mechanisms involved are described. High yields (~85%) of copper recovery were achieved. Besides, microorganism-mineral interactions were visualised by FISH and CLSM.

DETAILED COMMENTS:

  1. ln. 83, and also ln. 269, 278 306, 321: A. copahuensis should be in italic
  2. ln. 101: please correct mantaining to maintaining
  3. ln. 157: enzimatic should be enzymatic
  4. ln. 180: please correct diferent to different
  5. ln. 210-213: Chart (b) indicates that a vast majority of Fe was Fe(III) under all conditions (even condition 5 shows that at the end of the experiment around 800 ppm was Fe(II), which corresponds to 0.08%). Did such high Fe(III)/Fe(II) ratios affect redox potential in such significant ways (as indicated on ln. 197-201)?

6. Reaction 6: O2 – please use subscript for 2.

Author Response

We are very glad to submit the revised and corrected version of the manuscript Importance of initial interfacial steps during chalcopyrite bioleaching by a thermoacidophilic archaeon" to be considered for publication in Microorganisms. We would like to thank you for the thorough reading of the manuscript and their suggestions that contributed to improve its quality. In the corrected version we have addressed all the suggestions and comments (the modifications were highlighted in yellow and were made using the “track changes” function of MicrosoftWord). Please see the detailed responses to Reviewer #3 in the attached file.
